# Developing and validating a self-assessment tool for assessing confidence of nurse-midwives against competency domains of the International Confederation of Midwives, in India

**Bharati Sharma**[1], **Malin Bogren**[2]*, **Prabhu Ponnusamy**[3], **Vaibhav Rastogi**[3], **Minjan Patel**[1], **Sunanda Gupta**[4], **Arvind Pandey**[4†], **Ram Chahar**[4], **Emma Frances MCCONVILLE**[5], **Medha Gandhi**[6], **Paridhi Jha**[7]

1 Indian Institute of Public Health Gandhinagar, Gandhinagar, India, 2 University of Gothenburg, Gothenburg, Sweden, 3 Mamta, Health Institute of Mother and Child, New Delhi, India, 4 World Health Organization Country Office, New Delhi, India, 5 World Health Organization Headquarters, Geneva, Switzerland, 6 Bill and Melinda Gates Foundation, New Delhi, India, 7 Foundation for Research in Health Systems, Bangalore, India

† Deceased.
* malin.bogren@gu.se

**Data Availability Statement:** The nature of the data is such that the participants' identity can be

## Abstract

While midwives are recognized as primary care-providers for maternal and new-born care in many parts of the world, India is transitioning to professional midwifery. The pathway to midwifery in India has been through integrated nursing and midwifery education. Since 2018, India has introduced an 18-month post nursing midwifery education programme. To establish a baseline for midwives' competence and measure progress, there is a need for a quick, easy-to-administer and low-cost tool that can be used at scale to guide programme efforts. This paper describes the process of validation and testing of a tool to assess the confidence of practicing nurse-midwives as a proxy indicator for competence against the seven competency domains of the International Confederation of Midwives (2013). A total of 2198 nurse-midwives, providing maternal and new-born services at the time of the study, from 442 public and private hospitals across six states in India, responded to a self-administered questionnaire. The tool is psychometrically sound and can potentially be used in low-middle-income countries to measure confidence of midwives and nurse-midwives against international competence standards. It is suitable for quick large-scale programmatic assessments within a short time period, providing evidence to inform midwifery strengthening initiatives. The tool can be contextualized to suit specific country contexts. Since it was tested in India, where a distinct cadre of midwives is not yet established and nurse-midwives provide maternity care, the tool can be easily adapted for use in other countries in the South East Asia Region with similar contexts.

easily traced back on making available the raw data which creates serious ethical concerns. We will not be able to upload the raw data. However, the anonymized dataset can be made available upon a reasonable request via an email sent to the Chairman of IRB at FRHS, Bengaluru. email ID: irb@frhsindia.org. This, however, can make the data available for offline scrutiny in case of a reasonable query. Moreover, since this paper only describes the validation process of the tool, access to data is less relevant. The results of the survey in which the tool was used is not published yet since we await the approval of the Government of India.

**Funding:** This research was funded by the WHO country office in New Delhi, India (Grant Ref. No. 2018/862808-0) who were supported by the Bill and Melinda Gates Foundation (BMGF). The funding covered full or part salaries of all co-authors except SG, AP, RC, FEM, and MG. The authors RC, FEM from WHO and MG from BMGF contributed to the initial tool development, and provided comments on the draft manuscript.

**Competing interests:** After reading the policies for reporting competing interests of the journal, we declare that during the study implementation, preparation of the manuscript and its submission, RC worked at the WHO country office, FEM worked at WHO Head Quarters and MG worked at BMGF India Office.

## Introduction

Midwifery services provided by professional midwives, are an effective strategy for decreasing maternal and neonatal mortality and improving sexual, reproductive, maternal, new born and adolescent health [1–3]. Many counties, especially those in South East Asia with high maternal and neonatal mortality rates [4–6], face challenges such as the unavailability of professional midwives educated and regulated as per international standards [7–9], or incomplete integration of professional midwives into national health systems [10–12].

In 2018, the Indian government rolled out policy guidelines to establish midwife-led care [13], and introduced a curriculum for an 18- month post-nursing midwifery education according to the International Confederation of Midwives (ICM) recommendations [14]. Before this initiative and even now, India had staff nurses who graduated from two accredited integrated nursing and midwifery programmes: a 3-year Diploma in General Nursing and Midwifery (GNM) and a 4-year Bachelor's in Nursing (BSc Nursing). Both programmes include about six months of midwifery training. Graduates receive dual registration as Registered Nurses and Registered Midwives (RN & RM) and work as general staff-nurses. According to the 2018 policy, graduates from these programmes with an additional two years of clinical experience in maternity care are eligible for the new 18-month midwifery education programme to qualify as "Nurse practitioners in midwifery". In 2023, India introduced the National Nursing and Midwifery ACT [15] for legal protection of midwifery practice.

This paper describes the development and validation of a self-assessment tool measuring the confidence of practicing midwives against ICM midwifery competencies [16]. The tool was developed as a part of a larger study supported by World Health Organization (WHO), at the request of the Indian government. The government wished for a baseline assessment of competencies of staff nurses (referred to as nurse-midwives in this paper). This baseline was intended to inform the improvement of education standards in existing nursing and midwifery programmes and support the new midwife-led care policy. Although the government sought a competency assessment, we used confidence as a proxy for competence for reasons described in the next section.

Competence, as defined by the ICM, is the combination of knowledge, psychomotor, communication and decision-making skills that enable an individual to perform a specific task to a defined level of proficiency [7]. However, this conventional notion of competence overlooks the context in which performance occurs assuming that the professionals certified as competent can demonstrate uniform performance whatever the circumstances [17]. Straka (2004) discusses three components of competence: visible socio-culturally shaped external conditions, observable behaviour components, and non-observable cognitive, motivational, and emotional components [18]. This multifaceted nature of competence makes its objective measurement difficult [19].

The inclusive notion of competence is similar to, but not the same as, confidence. Confidence is influenced by situational, institutional, and dispositional factors [20–23]. For midwives, confidence means acquiring knowledge and psychomotor skills to a proficiency level and the ability to perform and persevere during difficulties and setbacks [24, 25]. This notion of confidence is "situated competence" [19]. As argued by Klass 2007 [17], competence when viewed from a performance orientation, reflects the situational relationships among providers, their patients, and the systems in which they work, making it partly dependent on the attributes of individual actors. Thus, confident handling of a situation is a principal criterion for evaluating situated competence. Therefore confidence could be a proxy for assessing fitness to practice or "situated competence" as confident midwives are likely to be competent midwives [26].

Considering the complexity in measuring competence, methods such as observations, vignettes and demonstrations on mannequins are time- and resource intensive and feasible only for small sample sizes. Examples include the Objective Structured Clinical Examination (OSCE) for measuring clinical competence of students [27, 28] and direct observation of clinical competence during labor, childbirth, and immediate postpartum care. Other studies have used vignettes and case scenarios to measure competence [29, 30]. Some studies have used vignettes [31], and case scenarios [32] to measure competence. The tool described in this paper offers a contextualizable, large- scale, cost-effective method for assessing confidence as a proxy for competence across all domains of ICM competencies.

## Methodology

### Study design, sites and sampling strategy

This study was a part of a larger cross-sectional research project to assess the confidence of midwifery teachers (those teaching midwifery at the time of the survey) and nurse-midwives from six selected states in India. Out of 36 states (8 union territories), six states were selected on their Maternal Mortality Ratios (MMR) of the years 2018 when the study was conceptualized. Knowing that nurse-midwives contribute to quality maternal and newborn services, we assumed they would have an impact on maternal and newborn mortality. These states were also chosen to ensure geographic representation across the country. The selected states were Assam (MMR, 237), Bihar (MMR, 165), Gujarat (MMR, 91), Karnataka (MMR, 108), Telangana (MMR, 81) and Uttar Pradesh (MMR, 201) [33].

A two-stage sampling strategy was adopted. At first stage, a sample of institutions was drawn offering Diploma and Degree programmes from the public and private sectors from a list of institutions provided by the State nursing councils, using probability proportionate to population size (PPS) sampling method. Since the study assessed confidence on competence of both midwifery teachers and practicing nurse-midwives, at the second stage, nurse-midwives from the clinical settings linked to each selected education institution were selected.

### Study participants

Nurse-midwives in this study were those general staff nurses who were posted in the maternity sections of health facilities at the time of the study. There were groups of nurse-midwives posted either to the antenatal, labour rooms or the postnatal wards including family planning for an indefinite period, after which they might get shifted within these wards or even to different specialities. Being posted in specific maternity sections meant that they did not perform full range of maternal and newborn services. However, all nurse midwives from maternity wards responded to all sections of the tool. A total of 2198 such staff-nurse-midwives from 442 hospitals/clinical sites participated in the study from six selected Indian states.

### Questionnaire development process

The study questionnaire has gone through several iterations. Sharma et al, 2018, had used four domains out of seven, of the ICM competencies (2013), antepartum, intrapartum, postpartum and new-born care to assess confidence of final year students in Gujarat state of India [34, 35]. The same tool was adapted and used in Sweden Kenya, Malawi, Tanzania, Uganda, Zambia, Zimbabwe, and Somaliland, for assessing confidence of final-year midwifery students [36–38]. The WHO HQ expanded the tool prepared by Sharma et al (2018) to include all seven competency domains of the ICM competencies. This tool from WHO was revised substantially and

adapted for the Indian context during which the co-authors from WHO Head Quarters as well as the WHO India office were involved.

The tool used the 2013 list of midwifery competencies given by the ICM [16], as the revised competencies were not published at the time the tool was being developed [9]. The tool includes all seven domains of ICM competencies required for graduating as a midwife. Questions were constructed for each skill statement under each of the seven domains. For each skill statement the respondents answered the question "*How confident are you to perform this skill* independently? For response the format was a 5-point Likert Scale *1) Not confident*, *2) Little confident*, *3) Somewhat confident, 4) Quite confident and 5) Very Confident*". Questions on background variables were added.

**Content validity.**   After simplification of each skill-statement, the content was checked for relevance to the Indian context, by a group of fifteen nurse-midwives and midwifery educators from the Society of Midwives of India (SOMI), experienced in teaching midwifery and supervising student's clinical practice. The skills in the ICM list which were out of scope of practice of nurse-midwives from India, were retained as the tool aimed to assess confidence against international standards, and a choice of *"Not relevant"* was added to the questionnaire. In addition, questions to capture the Indian context were added as general questions with yes and no answers for each competency domain, such as *"As part of your job, do you prescribe contraceptive methods"*?

**Face validity.**   For face validity, ten staff nurses responded to the questionnaire each from Gujarat and Karnataka. Revisions were made to ensure the right use of terminologies, simplicity and ease in understanding each statement. The questionnaire finally consisted of 230 skill statements across seven ICM competency domains. The questionnaire was translated into five Indian languages (Assamese, Gujarati, Hindi, Kannada and Telugu) and pilot tested again in each of the selected states participating in the study. A bilingual version with English and the regional language was used for data collection to facilitate quick understanding. The tool was transferred to the CAPI software and pilot tested again.

## Data collection

The data were collected during November 2019 to March 2020. A team of ninety data collectors and six supervisors were trained to ensure appropriate process of and quality of data collected across the six selected states. Data was collected on tablets using the computer assisted personal interview (CAPI) software; and a manual back-up was taken if required. The questionnaire could also be accessed through a link on the mobile phone. The participants had a choice to answer the English or the vernacular version of the questionnaire. It was mandatory to answer all questions. On an average the questionnaire took about 45–60 minutes to complete.

## Ethics approval and consent to participate

Ethical approval was given by the ethical review committee of the Foundation for Research in Health Systems (FRHS) Bangalore (IRB Reg. No.: IRB0009235). Permissions for participation of nurse-midwives from health facilities were granted by the Government of India and State Governments of each participating state. The Medical Superintendents of the surveyed health facilities gave permission to conduct the survey among staff-nurse-midwives in clinical facilities. Written informed consent was taken from each participant ensuring their confidentiality and voluntariness of participation.

## Evaluating psychometric properties

The Statistical package of Social Science (SPSS) version 24 (SPSS, Inc., Chicago, USA) was used to analyse and manage the data. The construct validity and reliability of the questionnaire

**Table 1. Results for sample adequacy and internal consistency of the midwifery practitioner questionnaire.**

| Competency Domain | KMO Test for Sample Adequacy | Bartlett's test of sphericity $\chi^2$(p-value) |
|---|---|---|
| **Competency-1:** Knowledge & skills from obstetrics, neonatology, social sciences, public health & ethics | 0.967 | 24324 (0.001) |
| **Competency-2:** Community health education & services-promote healthy family life, planned pregnancies & positive parenting | 0.942 | 46005 (0.001) |
| **Competency-3:** Aantenatal care, detection of complications & referral | 0.942 | 68589 (0.001) |
| **Competency-4:** Care during labour, conduct clean & safe birth & handle selected emergency situations | 0.942 | 88641 (0.001) |
| **Competency-5:** Comprehensive postpartum care | 0.942 | 55724 (0.001) |
| **Competency-6:** Comprehensive care of healthy infant from birth to 2 months | 0.942 | 33763 (0.001) |
| **Competency-7:** Abortion care within national laws/regulations & protocols | 0.942 | 16102 (0.001) |

was tested. Construct validity refers to how well a tool measures the construct it claims to measure. To do so a series of tests are required to 1) establish the suitability of the data to undertake construct validation; and 2) the tests for construct validation itself [39]. To establish the suitability of the data set, kaiser Meyer Olkin (KMO) test is performed to determine the factorability (adequacy of each variable and its completeness) of the data. The KMO value should be over 0.6 or higher for a newly constructed scale(40). The Bartlett's test of sphericity is performed to test whether the data is from random sample from a multivariate, normally distributed population which should be statistically significant (atleast <0.05) [40]. In this dataset, the KMO value was above 0.9 and Bartlett's test of sphericity reached statistical significance (<0.001), which supported the sample factorability and adequacy (Table 1).

Exploratory factor analysis (EFA) using Principle Component Analysis (PCA) is performed for a newly constructed scale to decrease redundancy and explore the dimensions of the construct under study. PCA helps to group items into subscales based on their correlation and usually each subscale describes a property of the construct under study. The number of retained subscale/domains was guided by the Kaisers criterion (eigenvalues >1), and also Cattell's scree test and inspection of the scree plot [40]. All the components with an eigenvalue 1 and statements loading above 0.40 were included. Finally, Cronbach alpha coefficients were calculated for each of the domains, to assess reliability.

## Results

### Participant's characteristics

**Background.** A total of 2198, nurse-midwives participated in the study from the six selected states (Table 2). The median age of the nurse-midwives was 31 years (range 20–70 years, SD 9.3). Majority were females (n = 2127, 97%), qualified as GNMs (n = 1634, 74%). The majority were working in the public health facilities (n = 1605, 73%), run by the government across the states. More than half of the nurse-midwives (n = 1015, 54%) had their posting in the labour room at the time of the survey.

**General clinical experience and experience in midwifery.** Three percent (n = 72) of the nurse-midwives did not have clinical experience of any kind, 39% (n = 869) had less than five years, 25% (n = 552) had between five to nine years of experience, while 32% (n = 705) had more than ten years of clinical experience. Nine percent (n = 202), of the nurse-midwives had

**Table 2. Background and professional characteristics of participants (N = 2198).**

| BACKGROUND/PROFESSIONAL CHARACTERISTICS | NUMBER (N = 2198) | % |
|---|---|---|
| AGE | | |
| <25 years | 308 | 14.0 |
| 25–29 years | 589 | 26.8 |
| 30–39 years | 803 | 36.5 |
| ≥40 years | 498 | 22.7 |
| SEX | | |
| Male | 67 | 3.0 |
| Female | 2 127 | 96.8 |
| Other | 4 | 0.2 |
| QUALIFICATION | | |
| Bachelors in nursing | 564 | 25.7 |
| Diploma in General Nursing & Midwifery | 1 634 | 74.3 |
| TYPE OF FACILITY | | |
| Public (Government) | 1 605 | 73.0 |
| Private | 593 | 27.0 |
| AREA OF POSTING IN FACILITY | | |
| Antenatal/OPD/ward | 356 | 18.8 |
| Labour Room | 1 015 | 53.5 |
| Post Natal ward | 469 | 24.7 |
| Family Planning section | 56 | 3.0 |
| TOTAL CLINICAL EXPERIENCE | | |
| None | 72 | 3.3 |
| <5 years | 869 | 39.5 |
| 5–9 years | 552 | 25.1 |
| ≥10 years | 705 | 32.1 |
| MIDWIFERY CLINICAL EXPERIENCE | | |
| No experience | 202 | 9.2 |
| <5 years | 1 253 | 57.0 |
| 5–9 years | 374 | 17.0 |
| ≥10 years | 369 | 16.8 |
| TOTAL BIRTHS ASSISTED IN PRECEDING 2 WEEKS OF SURVEY | | |
| Not posted at LR, no births assisted | 501 | 22.8 |
| Posted at LR, assisted up to six births | 671 | 30.5 |
| Posted at LR, assisted seven or more births | 1 026 | 46.7 |
| CONTRACEPTIVE SERVICES | | |
| Not allowed to prescribe contraceptives | 646 | 29.4 |
| Prescribe contraceptives including Intra Uterine Device/mechanical | 353 | 16.1 |
| Prescribe contraceptive excluding Intra Uterine Device/mechanical | 1 199 | 54.5 |

no clinical experience in midwifery, 57% (n = 1253) had less than five years of experience, while 35% (n = 743) had more than five years of experience.

As seen in Table 2, almost 77% of the participants were in practice of assisting births as measured by "births assisted in preceding two weeks of survey"; 30% (n = 671) having assisted up to six births and 47% (n = 1026) having assisted more than seven births in the last two weeks of survey.

As seen in Table 3, for competency-1, PCA yielded three factors from a total of 26 items; for competency-2, five factors from 32 items; for competency-3 two factors from 50 items; for

**Table 3. Psychometric properties from PCA for seven competencies for midwifery practitioners.**

| | Items | Item Analysis | | | Construct validity (factor analysis) | | | Internal Reliability Cronbach's α |
|---|---|---|---|---|---|---|---|---|
| | | Item-total correlation range | Item-Subscale Correlation Range | Subscale-Total scale correlation | Eigen value | % Explained Variance | Loading Range | |
| **Competency-1:** Knowledge & skills from obstetrics, neonatology, social sciences, public health & ethics (26 items) | | | | | | | | |
| Factor 1: Incorporate sociocultural aspects in caregiving | 10 | 0.55, 0.65 | 0.64, 0.71 | 0.89 | 12.68 | 48.79 | (0.5, 0.88) | 0.873 |
| Factor 2: Identify cases of domestic violence and other sexual abuse | 4 | 0.57, 0.62 | 0.65, 0.76 | 0.74 | 1.74 | 6.69 | 0.68, 0.83 | 0.831 |
| Factor 3: Handle instruments, interpret, maintain medical records and legal framework for midwifery | 12 | 0.54, 0.69 | 0.61, 0.71 | 0.93 | 1.33 | 5.11 | -0.78, -0.45 | 0.895 |
| **Competency-2:** Community health education & services- promote healthy family life, planned pregnancies & positive parenting (32 items) | | | | | | | | |
| Factor 1: Prenatal health education, disease prevention and counselling | 10 | 0.5, 0.62 | 0.61, 0.77 | 0.72 | 12.28 | 38.39 | 0.47, 0.85 | 0.913 |
| Factor 2: Perform and interpret findings of gynaecological screening tests | 7 | 0.54, 0.56 | 0.52, 0.8 | 0.38 | 4.11 | 12.84 | 0.45, 0.90 | 0.916 |
| Factor 3: Insert and remove female contraceptive devices | 4 | 0.47, 0.56 | 0.7, 0.8 | 0.51 | 1.75 | 5.47 | -0.91, -0.78 | 0.891 |
| Factor 4: Take and interpretive reproductive history | 6 | 0.48, 0.61 | 0.61, 0.76 | 0.66 | 1.33 | 4.16 | 0.45, 0.83 | 0.861 |
| Factor 5: Conduct & interpret general physical examination and laboratory | 4 | 0.53, 0.62 | 0.67, 078 | 0.69 | 1.14 | 3.57 | 0.42, 0.66 | 0.834 |
| **Competency-3:** Aantenatal care, detection of complications & referral (50 items) | | | | | | | | |
| Factor 1: Counselling for Birth Preparedness complication readiness | 13 | 0.55, 0.7 | 0.61, 0.77 | 0.91 | 21.06 | 42.11 | 0.49, 0.79 | 0.918 |
| Factor 2: Perform screening examinations to identify deviations | 3 | 0.43, 0.51 | 0.63, 0.7 | 0.58 | 2.57 | 5.03 | 0.48, 0.76 | 0.733 |
| Factor 3: Identify signs of and confirm pregnancy | 6 | 0.49, 0.7 | 0.6, 0.72 | 0.79 | 2.09 | 4.17 | 0.41, 0.71 | 0.789 |
| Factor 4: Assess and counsel for maternal nutrition | 4 | 0.58, 0.66 | 0.65, 0.72 | 0.86 | 1.44 | 2.87 | 0.41, 0.47 | 0.862 |
| Factor 5: Identify, manage and counsel for antenatal complications | 9 | 0.63, 0.68 | 0.72, 0.76 | 0.88 | 1.31 | 2.61 | 0.54, 0.74 | 0.918 |
| Factor 6: Conduct and interpret antenatal examinations | 11 | 0.59, 0.67 | 0.62, 0.73 | 0.89 | 1.23 | 2.47 | -0.84, -0.50 | 0.923 |
| **Competency-4:** Care during labour, conduct clean & safe birth & handle selected emergency situations (55 items) | | | | | | | | |
| Factor 1: Managing 2nd and 3rd stages of labour including complications | 21 | 0.53, 0.7 | 0.6, 0.71 | 0.94 | 24.71 | 44.92 | 0.42, 0.75 | 0.955 |
| Factor 2: Providing respectful maternal and neonatal care during labour and birth | 8 | 0.57, 0.65 | 0.78, 0.82 | 0.79 | 3.59 | 6.52 | 0.73, 0.83 | 0.939 |
| Factor 3: Identifying and performing non-penetrative management of shock during labour | 5 | 0.62, 0.66 | 0.81, 0.85 | 0.77 | 1.91 | 3.47 | 0.54, 0.91 | 0.919 |
| Factor 4: Managing special cases (e.g., HIV in pregnancy) and referral | 5 | 0.55, 0.65 | 0.66, 0.73 | 0.79 | 1.74 | 3.17 | 0.51, 0.69 | 0.826 |
| Factor 5: Preparing labour room and managing the 1st stage of labour | 14 | 0.57, 0.69 | 0.67, 0.74 | 0.92 | 1.35 | 2.45 | 0.44, 0.71 | 0.938 |
| Factor 6: Performing IV management of shock | 2 | 0.45, 0.46 | 0.82, 0.86 | 0.61 | 1.05 | 1.91 | 0.41, 0.52 | 0.742 |
| **Competency-5:** Comprehensive postpartum care (33 items) | | | | | | | | |

*(Continued)*

**Table 3.** (Continued)

| | Items | Item Analysis | | | Construct validity (factor analysis) | | | Internal Reliability Cronbach's α |
|---|---|---|---|---|---|---|---|---|
| | | Item-total correlation range | Item-Subscale Correlation Range | Subscale-Total scale correlation | Eigen value | % Explained Variance | Loading Range | |
| Factor 1: Performing postpartum examinations and identifying complications | 10 | 0.62, 0.7 | 0.73, 0.78 | 0.91 | 16.85 | 51.06 | 0.44, 0.88 | 0.927 |
| Factor 2: Support breastfeeding and counselling for newborn/maternal loss | 7 | 0.53, 0.71 | 0.69, 0.80 | 0.82 | 2.39 | 7.23 | 0.47, 0.7 | 0.901 |
| Factor 3: Identifying and managing late postpartum complications and referral | 8 | 0.55, 0.68 | 0.66, 0.79 | 0.89 | 1.45 | 4.4 | 0.47, 0.74 | 0.910 |
| Factor 4: Providing postnatal counselling to women and families | 8 | 0.65, 0.72 | 0.74, 0.8 | 0.91 | 1.01 | 3.06 | -0.83, -0.47 | 0.923 |
| Competency-6: Comprehensive care of healthy infant from birth to 2 months (23 items) | | | | | | | | |
| Factor 1: Managing and referral for high-risk newborn babies and counselling couples on newborn care | 12 | 0.46, 0.61 | 0.53, 0.76 | 0.83 | 12.26 | 53.31 | 0.4, 0.79 | 0.923 |
| Factor 2: Providing immediate newborn care as per guidelines | 11 | 0.56, 0.65 | 0.69, 0.75 | 0.81 | 1.2 | 5.21 | -0.88, -0.47 | 0.931 |
| Competency-7: Abortion care within national laws/regulations & protocols (11 items) | | | | | | | | |
| Factor 1: Providing comprehensive postpartum care | 11 | 0.56, 0.74 | 0.57, 0.68 | NA | 6.54 | 59.45 | NA | 0.929 |

competency-4, two factors from 55 items; for competency-5, four factors from 33 items; for competency-6, two factors from 23 items; and for competency-7 only one factor was identified.

Overall, all the subscales/factors were unidimensional except a few as described in Table 4. The decisions to change items which loaded on more than one factor or those which did not load, were made based on the country context and the best fit. These decisions did not have a major effect on the internal consistency of the tool, in terms of the subscale to total and item to subscale correlations (Table 3).

## Discussion

This paper describes the process of validating and testing a tool to measure confidence of nurse-midwives in India, against international standards of competence. The tool consists of 230 items (skill statements) across the seven ICM competency domains. The study involved 2198 nurse-midwives from the maternity sections of public and private health facilities in six selected states of India.

The sample size was adequate according to the recommended size needed for factorability of data [40]. Using ICM competency list adds to tool's validity, as the ICM list of competency followed scientific rigor, with representation of experts in high and low resource settings, and various types of midwifery programmes globally [41]. This tool was developed before the ICM revised the competency list in 2019 [42]. Apart from general competencies related to professionalism and management, the essential competencies in both the 2013 and the current one, are similar but organized under different headings. Therefore, the tool remains relevant.

The tool demonstrates good internal consistency, evidenced by the correlation of subscales identified through factor analysis with total scores and scores for each competency domain. It

**Table 4. Details of decisions regarding items which double loaded or did not load.**

| Competency domain | PCA results | Decisions taken |
|---|---|---|
| **Competency-1:** Knowledge & skills from obstetrics, neonatology, social sciences, public health & ethics | 3 factors extracted. | Retained as it is |
| **Competency-2:** Community health education & services- promote healthy family life, planned pregnancies & positive parenting | 5 factors extracted. One item "*Take accurate preventive measures in case of exposure to HIV for myself and co-workers*" loaded on factor-2 as well as factor-3 | Retained under factor-3 because of better fit. |
| **Competency-3:** Aantenatal care, detection of complications & referral | 6. Factors extracted. Four items did not load; • *Assess maternal nutrition through Pallor* • *Help the woman in developing a culturally sensitive and affordable diet plan* • *Use the Doppler to record fetal heart sounds* • *Identify deviation from normal for ALL the above-mentioned parameters* | Included under factor-2 considering the country context of high prevalence of Anaemia, and heterogeneity in cultures • *Assess maternal nutrition through Pallor* • *Help the woman in developing a culturally sensitive and affordable diet plan* Included into factor-3 • *Use the Doppler to record fetal heart sounds* • *Identify deviation from normal for ALL the above-mentioned parameters* Item *Diagnose an ectopic pregnancy* moved from factor- 4 to factor-1Item *Assess maternal nutrition through: Weight and weight gain* &*Advise woman on correct nutritional intake* moved from factor-1 to factor-3, based on better fit |
| **Competency-4:** Care during labour, conduct clean & safe birth & handle selected emergency situations | Six factors extractedThree Items did not load • *D 28b I can provide a safe environment for mother and infant to promote attachment (bonding)* • *D33b I can administer prescribed drugs or drugs as per the national protocols and guidelines* • *D35b I can perform aortic compression* | D 28b, 33b, 35b included in factor-2 Item *D41b I can insert intravenous line* moved from factor-6 to factor-3 because of better fit |
| **Competency-5:** Comprehensive postpartum care | Four factors extracted. | Item *E25b I can counsel a family after a maternal loss* moved from factor-4 to factor-3 due to better fit |
| **Competency-6:** Comprehensive care of healthy infant from birth to 2 months | Two factors extracted. | Retained as it is |
| **Competency-7:** Abortion care within national laws/regulations & protocols | One factor extracted. | Retained as it is |

also shows good validity and reliability. However, further research using more rigorous statistical methods such as structural equation modelling, is needed to confirm these findings [43].

There are tools tested in developed countries, either self-assessment of competence [35] or assessment by supervisors in case of students [30]. Some tools are based on national standards [30], while others use the ICM list of competencies [16, 20]. Some tools focus on assessing a subset of midwifery competencies, such as antenatal care, intra-partum care, cultural competency, rather than holistic assessment of all the seven ICM competencies, that is, 1) multi-disciplinary cultural competence, 2) pre-pregnancy counselling and education for happy parenthood, 3) antepartum care, 4) intrapartum care, 5) postpartum care, 6) new born care, and 7) abortion care. Other studies focus domains of competence such as respectful maternal care [44], decision making skills of midwives [45], cultural competence [46] or spiritual competence [47].

The tool described in this paper can provide quick, large-scale baseline assessments to inform programmes and policies by identifying specific domains and skills that need strengthening particularly in countries transitioning to midwifery model of care. It can help programme managers and educators design priority-based pre-service and in-service midwifery education, optimizing resources in low-resource settings. By breaking down each competency domain into detailed list of skills, the tool makes the assessment comprehensive and easy for stakeholders to understand. For instance, our participants reported low confidence for

intrapartum care, expressing moderate to low confidence in 41out of 55 total skill statements in the intrapartum domain. Many nurse-midwives also expressed a lack of confidence in dealing with cases of rape, and domestic violence. These skill-gaps can directly contribute to improving the quality of midwifery education.

Another strength of the tool is its ease of adaption to any country context. Additional questions about types of programmes and scope of practice can be easily contextualized to local situations.

The researchers decided to measure confidence as a proxy for competence for this tool, as confidence mediates between competence and performance. Performance is "Situated competence (grounded in the situation)" [48], similar to Butler's concept [25] of integrated competence" where a competent midwife is the one who can perform safe practice in all situations. Since nurse-midwives in India are not educated to the level of international standards of midwifery, and also have circumstance driven scope of practice [49], their competence would have been highly influenced by their context. Therefore, it was assumed that measuring confidence would be a better reflection of their "situated competence". This makes the tool relevant for countries of South East Asia which share a similar if not the same context related to nursing and midwifery.

## Limitations

Since the study assessed the confidence in competence of both midwifery teachers and practicing nurse-midwives, nurse-midwives were selected from clinical settings linked to each education institution included in the study for convenience. Consequently, there are higher numbers of medical college hospitals and district hospitals included in the study compared to community health centres and the primary health centres. There are fewer nurse-midwives posted at the community health centres and primary health centres compared to the large hospitals, as the majority of the childbirth workload being handled by these larger hospitals. Therefore, measuring confidence at these higher facilities where the nurse-midwives have more opportunities to practice their skills would yield results on the higher side. This can be viewed as a limitation of the study. However, the Bartletts test of sphericity which measures the representativeness of the population, revealed our sample to be random and representative.

As the current tool relies on self-assessment, there is a possibility of espondents providing socially desirable responses. This concern was partly addressed by competent researchers who provided effective orientation to respondents. The structure of the current tool follows the organization of the ICM list of competencies, which are divided into seven domains, each representing wide specialized areas. This makes the tool more suitable for baseline assessments for midwifery strengthening programmes rather than precise individual assessments of competence. For more precise assessments of competence in such important but complex areas, other methods or tools can be used alongside this tool. As Normana et al (2002) also concluded, no single tool can adequately measure competence, especially clinical competence [50]. Therefore, other methods such as direct observations can be carried out on a subset of participants answering this tool.

## Conclusions

The tool described in this paper to measure the confidence of nurse-midwives is valid and reliable in its current form, as it is psychometrically tested and is based on ICM list of competencies. The tool could prove to be valuable for conducting robust large-scale assessments of confidence among practicing nurse-midwives/midwives against international standards of

competence. Such assessments could help in establishing baseline for strengthening pre-service and in-service education of midwives.

## Supporting information

**S1 Checklist. Inclusivity in global research.**
(DOCX)

**S1 File. Copy of the tool for background information for participants.**
(PDF)

**S2 File. Copy of the tool for measuring confidence of midwives against ICM competencies.**
(PDF)

## Acknowledgments

We would like to express our deepest gratitude to Dr. Pandey, who passed away before the submission of the final version of this manuscript, for his invaluable contributions to the study. Our sincere thanks go to the study respondents for their enthusiastic participation, and to the local clinical sites for assisting in mobilizing respondents. We are grateful to the Ministry of Health and Family Welfare (MoHFW), Government of India, and the State Health Departments of the six states for their valuable suggestions towards the study design and for facilitating the study with necessary approvals and communication with educational and clinical facilities. We also extend our thanks to the panel of experts, including representatives from the Society of Midwives of India, the State Nursing Councils, the Indian Nursing Council, and the Nursing Division at MoHFW, for their role in the content validation of the tool. We acknowledge the dedication of the research team members who adhered to the study protocol and ensured data quality, despite the challenges of the COVID-19 pandemic.

## Author Contributions

**Conceptualization:** Bharati Sharma, Sunanda Gupta, Ram Chahar, Emma Frances MCCONVILLE, Paridhi Jha.

**Data curation:** Arvind Pandey.

**Formal analysis:** Bharati Sharma, Prabhu Ponnusamy, Vaibhav Rastogi, Minjan Patel, Arvind Pandey, Paridhi Jha.

**Investigation:** Minjan Patel.

**Methodology:** Bharati Sharma, Prabhu Ponnusamy, Vaibhav Rastogi, Sunanda Gupta, Arvind Pandey, Emma Frances MCCONVILLE, Medha Gandhi, Paridhi Jha.

**Project administration:** Prabhu Ponnusamy, Vaibhav Rastogi, Minjan Patel, Sunanda Gupta.

**Software:** Prabhu Ponnusamy.

**Supervision:** Vaibhav Rastogi, Minjan Patel, Sunanda Gupta, Ram Chahar, Paridhi Jha.

**Validation:** Bharati Sharma, Minjan Patel, Arvind Pandey, Paridhi Jha.

**Writing – original draft:** Bharati Sharma, Malin Bogren.

**Writing – review & editing:** Malin Bogren, Vaibhav Rastogi, Minjan Patel, Sunanda Gupta, Ram Chahar, Emma Frances MCCONVILLE, Medha Gandhi, Paridhi Jha.

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
