## [Decision Letter · Decision Letter 0]

12 Apr 2024

PGPH-D-23-02310

Developing and validating a self-assessment tool for assessing confidence of Indian Nurse-Midwives on competency domains of the International Confederation of Midwives

Dear Dr. Sharma,

Thank you for submitting your manuscript to PLOS Global Public Health. After careful consideration, we feel that it has merit but does not fully meet PLOS Global Public Health’s publication criteria as it currently stands. Therefore, we invite you to submit a revised version of the manuscript that addresses the points raised during the review process.

The manuscript has been evaluated by two reviewers, and their comments are available below. The reviewers have raised a number of concerns that need attention. In particular they request revisions to more clearly define the contribution of this study, as well as several clarifications to the context and methodology. Could you please revise the manuscript to carefully address the concerns raised?

We look forward to receiving your revised manuscript.

Kind regards,

Marianne Clemence

Staff Editor

Journal Requirements:

2. Please send a completed 'Competing Interests' statement, including any COIs declared by your co-authors. If you have no competing interests to declare, please state "The authors have declared that no competing interests exist". Otherwise please declare all competing interests beginning with the statement "I have read the journal's policy and the authors of this manuscript have the following competing interests:"

3. Please amend your detailed Financial Disclosure statement. This is published with the article. It must therefore be completed in full sentences and contain the exact wording you wish to be published.

If you did not receive any funding for this study, please simply state: “The authors received no specific funding for this work.

4. We have noticed that you have uploaded Supporting Information files, but you have not included a list of legends. Please add a full list of legends for your Supporting Information files after the references list.

5. In the online submission form, you indicated that "The nature of the data is such that the participants' identity can be easily traced back on making available the raw data which creates serious ethical concerns. We will not be able to upload the raw data, however, can make it available for offline scrutiny at the corresponding author's office in case of a reasonable query". All PLOS journals now require all data underlying the findings described in their manuscript to be freely available to other researchers, either 1. In a public repository, 2. Within the manuscript itself, or 3. Uploaded as supplementary information.

Additional Editor Comments (if provided):

Reviewers' comments:

Reviewer's Responses to Questions

**Comments to the Author**

1. Does this manuscript meet PLOS Global Public Health’s publication criteria? Is the manuscript technically sound, and do the data support the conclusions? The manuscript must describe methodologically and ethically rigorous research with conclusions that are appropriately drawn based on the data presented.

Reviewer #1: Yes

Reviewer #2: Yes

2. Has the statistical analysis been performed appropriately and rigorously?

Reviewer #1: Yes

Reviewer #2: Yes

3. Have the authors made all data underlying the findings in their manuscript fully available (please refer to the Data Availability Statement at the start of the manuscript PDF file)?

Reviewer #1: Yes

Reviewer #2: Yes

4. Is the manuscript presented in an intelligible fashion and written in standard English?

Reviewer #1: Yes

Reviewer #2: Yes

5. Review Comments to the Author

Reviewer #1: Thank you for giving me the opportunity to review this paper.

I thought that the technical content of the paper, around the description of the tool and its validation – the factor analysis etc. – was very strong. I didn’t have any technical comments on it.

However, I think the up front framing of the article and its contribution needs more clarity and justification.

My understanding is that you are arguing that:

- We want to measure competence, but this is expensive/hard to do at scale/hard to do often through traditional measures

- Instead, we can measure self-assessed confidence – that this is a good ‘proxy’ for competence and it is feasible to do this cheaply and at scale

- This has been done previously for a subset of the ICM domains; the contribution of this tool is that it covers all ICM domains (I may have misunderstood this)

- The findings show that it works as a tool, and can be contextualized

- The contribution is therefore to offer a contextualizable tool for large scale, cost-effective assessments of confidence that proxy for competence

This would be a clear justification (if I have understood it correctly). However, I don’t feel you set this out clearly in the beginning.

The background second para (line 64 onwards) doesn’t convincingly make a case as to why we cannot rely on existing approaches to measure competence for certain needs (e.g., large scale assessments) – which you do well starting line 283 but that is far too late. I would have expected you to set out clearly what the problem is, that this article is seeking to solve. Without this, it is not clear why you start talking about confidence when the background is all about attempts to measure competence. A clear sentence or para that says: this is what the article does, and this is what it contributes, would be very helpful.

I would considering keeping the section on competence vs confidence to follow on directly from this section (after line 79), and then bring the Indian context after. At the moment, the Indian context section is in the middle of the two sections that provide the conceptual justification, which breaks the flow.

On the competence vs confidence section, this would have a clearer narrative flow if you had established that we cannot measure competence easily at scale using traditional measures, so we need to measure confidence instead. I found this section a little weak – I would have expected a stronger theoretical and empirical justification as to why measuring confidence is a good proxy for measuring competence. I would have brought in some examples of where it has been successfully used elsewhere here.

On the methodology, I got a little bit confused about this sentence: “At the second stage, nurse-midwives were randomly selected from the clinical settings linked to each selected education institution” (line 145). Do you mean you got a list of students who had graduated from the selected teaching institutions, and went and found them wherever they were working? Or they are only from clinical settings attached to the education institution, and not other facilities across the states? In which case how representative are they? This needs more clarification and explanation.

On the evaluating section (line 210), it would be helpful if you set out: to test the validity of the tool, we need to test whether it has good internal consistency, reliability etc. etc. And to do this, we do (e.g., factor analysis). You jump straight into the weeds of how you do these tests without setting out why these are the right tests to demonstrate that the tool works. For a non-technical reader, some clearer framing would be helpful.

In the results section – line 235 – is the 75% figure right?

I think if you set up the purpose of the article more clearly, you would also be able to reshape the discussion into a clearer narrative. And the abstract also.

I did wonder whether you wanted to present some of the findings of the tool to justify its utility (not just its validity). For example, you say that ‘another strength of this validated tool is that it enables a quick and efficient way to identify skill gaps’ – do you have some examples of this? What did the findings enable in terms of policy or programmatic change?

For the external validity point, perhaps you could reflect on the process you followed for India (content and face validity) and what it would take to tailor this tool to a different context?

Reviewer #2: Comments on PGPH-D-23-02310

Developing and validating a self-assessment tool for assessing confidence of Indian

Nurse-Midwives on competency domains of the International Confederation of

Midwives

General

This is a useful study which should be publishable following some minor amendments and clarifications.

Title long version

Suggest editing to: Developing and validating a self-assessment tool for assessing confidence of Nurse-Midwives on competency domains of the International Confederation of Midwives in India

Abstract

Line 30-37

This looks like a clear summary. I suggest adding a brief explainer /definition of nurse-midwife. Those roles should not be conflated, the abstract starts with reference to midwives and midwifery care and midwifery cadre, then jumps to nurse-midwife. E.g transition to professional midwifery from ?

Line 43

There is no evidence of transferability to high-income countries, please provide further explanations on how this can be applied to high-income when setting is low-middle income countries.

Background

Line 80 onwards

Please provide information on educational pathways /degree level of the nursing programme too, as there is no context of the add-on 18 months. Are those programmes provided by higher education institutions- please add brief contextual explainer.

Line 130

Please expand on the concept of ‘situated competence’.

Methodology

Line 138

Please clarify why chosen for mortality ratio, it is unclear.

Line 151

Do labour rooms refer to birth rooms? Or nurse-midwife only provide labour care. Please clarify: expand or edit.

Line 158

Please remove first author reference, irrelevant here. Obvious from author list who is the 1st author. Please describe iterations instead and how team members worked together, removed/minimised bias and agree or found resolution when in disagreement.

Line 161

Please remove African countries: clarify which countries instead

Line 162

Again the reference of first author having authored other papers is completely irrelevant for the purpose of the paper, remove.

Line 177

Refer to nurse-midwives in India and not ‘indian’

Results

Please report absolute numbers throughout not just percentages. E.g 97% females add (n. x, 97%).

Line 229

Please expand on participants age. It is unclear how they can be qualified nurse-midwife at 18 years old.

Line 231

Please give context to employment/employer status (e.g public health facilities)

Table 2 and Line 235

Please clarify timeframe of birth attendance (how many births in how long/throughout their career or last 2 weeks as above).

Discussion

Please amend the discussion to include reference to existing literature, what is new and how it compares with published literature.

Line 308

Please expand comparison to other existing tools

6. PLOS authors have the option to publish the peer review history of their article (what does this mean?). If published, this will include your full peer review and any attached files.

**Do you want your identity to be public for this peer review?** For information about this choice, including consent withdrawal, please see our Privacy Policy.

Reviewer #1: **Yes: **Thomas Newton-Lewis

Reviewer #2: No

---

## [Editor Report · Decision Letter 1]

6 Sep 2024

REVISED TITLE. Developing and validating a self-assessment tool for assessing confidence of Nurse-Midwives against competency domains of the International Confederation of Midwives in India

PGPH-D-23-02310R1

Dear Dr Bogren,

We are pleased to inform you that your manuscript 'REVISED TITLE. Developing and validating a self-assessment tool for assessing confidence of Nurse-Midwives against competency domains of the International Confederation of Midwives in India' has been provisionally accepted for publication in PLOS Global Public Health.

Best regards,

Laila Akbar Ladak, PhD, MScN, BScN, RN

Section Editor